



Atmospheric
Chemistry
and Physics

# Modelling mixed-phase clouds with the large-eddy model UCLALES–SALSA

**Jaakko Ahola**[1], **Hannele Korhonen**[1], **Juha Tonttila**[2], **Sami Romakkaniemi**[2], **Harri Kokkola**[2], **and Tomi Raatikainen**[1]

[1]Finnish Meteorological Institute, Helsinki, Finland
[2]Finnish Meteorological Institute, Kuopio, Finland

**Correspondence:** Jaakko Ahola (jaakko.ahola@fmi.fi)

**Abstract.** The large-eddy model UCLALES–SALSA, with an exceptionally detailed aerosol description for both aerosol number and chemical composition, has been extended for ice and mixed-phase clouds. Comparison to a previous mixed-phase cloud model intercomparison study confirmed the accuracy of newly implemented ice microphysics. A further simulation with a heterogeneous ice nucleation scheme, in which ice-nucleating particles (INPs) are also a prognostic variable, captured the typical layered structure of Arctic mid-altitude mixed-phase cloud: a liquid layer near cloud top and ice within and below the liquid layer. In addition, the simulation showed a realistic freezing rate of droplets within the vertical cloud structure. The represented detailed sectional ice microphysics with prognostic aerosols is crucially important in reproducing mixed-phase clouds.

## 1 Introduction

Clouds are known to have a prominent influence on the hydrological cycle and the atmospheric radiation balance. While significant advances have been made in characterisation of liquid-phase clouds, the microphysical processes, especially heterogeneous ice nucleation, dynamics and radiative effects of mixed-phase and ice clouds remain more poorly constrained. This is mainly because of challenges in obtaining representative observations and a lack of a detailed enough representation of microphysics in climate and numerical weather prediction models. Specific challenges are known to be associated with aerosol–cloud interactions (Cox, 1971; Knight and Heymsfield, 1983; Curry, 1995; Solomon et al., 2007; Stevens and Feingold, 2009; Morrison et al., 2011a; Morrison, 2012; Li et al., 2013).

What we know about mixed-phase clouds is that by definition supercooled liquid droplets co-exist with ice crystals. Such clouds are frequent at temperatures between $-10$ and $-25\,°C$ (Filioglou et al., 2019) but can be present from $-35$ to $0\,°C$ and require specific microphysical and dynamical conditions (Andronache, 2017). Ice crystals can form either by homogeneous or heterogeneous freezing (the term *nucleation* used also). At temperatures lower than $-38\,°C$, liquid droplets can freeze homogeneously without the need for ice-nucleating particles (INPs). In heterogeneous ice nucleation, freezing initiates from the surface of seed particles and can occur at higher temperatures than homogeneous ice nucleation. Droplet freezing processes are not yet fully understood and quantified despite of decades of research (Phillips et al., 2008; Atkinson et al., 2013; DeMott et al., 2011). Kiselev et al. (2017) stated that ice formation on aerosol particles (heterogeneous ice nucleation) is a process of crucial importance to Earth's climate, but it is not understood at the molecular level. However, in Morrison et al. (2011a) it is noted that although many details of droplet freezing are poorly understood, enough knowledge exists to draw first-order (ice water path) conclusions. Furthermore, droplet freezing models and even the representation of cloud structure often require a resolution that is too detailed for large-scale models. For instance, the structure of Arctic and mid-altitude clouds is complex, with a layered structure with liquid near cloud top and ice within and below the liquid layer (Curry et al., 1997; Hobbs and Rangno, 1998; Pinto, 1998; Rangno and Hobbs, 2001; Zuidema et al., 2005; Shupe et al., 2006; Verlinde et al., 2007; de Boer et al., 2009; McFarquhar et al., 2011; Mor-

rison et al., 2011a). The lack of a proper calculation of ice processes in climate models is seen in comparisons to observations of mid- and high-latitude mixed-phase clouds. These models tend to underestimate the lifetime of such clouds (Andronache, 2017). Better quantification of droplet freezing processes is expected to narrow the gap between observations and model results.

Including a detailed aerosol description is vital in cloud-resolving models. Scarcity of INPs is an important factor in mixed-phase cloud lifetime and structure, since roughly one in a million particles acts as an ice nucleus, and even these particles might have highly different ice-forming activity at different temperatures (Lebo et al., 2008; Morrison et al., 2011a). Therefore, the loss of INPs along with precipitating ice crystals limits cloud glaciation and dissipation (Rauber and Tokay, 1991; Harrington et al., 1999; Avramov and Harrington, 2010). Describing this process is not possible without a detailed description of aerosols, as is demonstrated in a 1-D cloud model study by Morrison et al. (2005). The significance of aerosols is shown in Filioglou et al. (2019) wherein a high aerosol load was linked with a higher occurrence of mixed-phase clouds. Also, the Norgren et al. (2018) study shows that there is less ice in polluted clouds. Andronache (2017) and Morrison et al. (2011a) provide comprehensive review resources for further details about mixed-phase clouds.

There is a growing number of studies focusing on examining the properties of mixed-phase or ice clouds by combining observations and models, including large-eddy simulation (LES) modelling and other cloud-resolving models (CRMs) (Jiang et al., 2000; Klein et al., 2009; Morrison et al., 2011b; Ovchinnikov et al., 2014; Andronache, 2017). Large-eddy simulations are particularly attractive for modelling boundary layer clouds since they offer a good compromise between computational cost and accuracy in terms of model resolution (Tonttila et al., 2017; Andronache, 2017). LES models explicitly solve the largest eddies in turbulent flows and use parameterisations for the smallest length scales. In atmospheric applications they are usually coupled with cloud microphysical packages. Recent developments in the computational performance of supercomputers have also made an explicit and detailed description of aerosol–cloud–ice microphysical interactions possible in LES modelling, allowing for the investigation of non-linear cloud phenomena, such as secondary ice production and heterogeneous ice nucleation.

There are several LES models that solve cloud-related interactions (Fridlind et al., 2012; Khain et al., 2004; Savre and Ekman, 2015; Fu and Xue, 2017). In comparison to those models, we present an LES model, UCLALES–SALSA, that brings additional value with a more detailed aerosol description. UCLALES–SALSA explicitly resolves interactions between aerosols, ice crystals and cloud droplets with sectional microphysics for all hydrometeors while keeping track of the aerosol dry size distribution. The sectional description, especially for aerosols, is a clear asset of UCLALES–SALSA and

we have now also extended this description for ice crystals. This sectional aerosol description allowed the implementation of a detailed heterogeneous freezing processes. First, the model results are compared with previously published modelling results. Finally, we demonstrate the benefits of this approach to handle heterogeneous freezing over more simplified aerosol–ice–cloud treatments.

## 2 Model description

The UCLALES–SALSA (Tonttila et al., 2017) model consists of two components: first, the widely used large-eddy simulator UCLALES (Stevens et al., 1999, 2005), and second, the aerosol bin microphysics model SALSA (Sectional Aerosol module for Large-Scale Applications) (Kokkola et al., 2008; Tonttila et al., 2017; Kokkola et al., 2018). UCLALES handles e.g. surface fluxes, transportation of microphysical prognostic variables and atmospheric dynamics including turbulence. The previous version of UCLALES–SALSA incorporated interactions between aerosols, clouds and drizzle (Tonttila et al., 2017). Now we have extended the model with a description for ice crystals. In this study, we focus on how ice crystals and ice-nucleating particles (INPs) interact with clouds while tracking sectional aerosol size distribution.

Figure 1 illustrates the microphysical treatment of different hydrometeor classes and their size distributions in UCLALES–SALSA. All four classes (aerosol, cloud and rain droplets, and ice crystals) are tracked with a bin scheme. The bin scheme offers the benefit of greater accuracy in simulating interactions between hydrometeors of different sizes. Better accuracy is gained by dividing the size distribution into bins. This also enables better flexibility as the shape of the distribution is allowed to evolve. Bulk schemes provide a simpler method and track one or several moments of the size distribution, whereby the shape of the distribution is prescribed. The disadvantage of the bin scheme is higher computational cost compared to the bulk scheme.

Three of the hydrometeor classes, i.e. aerosol, cloud droplets and ice, are further divided into parallel bins labelled a and b as shown in Fig. 1. This division into a and b bins is done to enable the tracking of externally mixed distributions and to see how different particles affect clouds. For aerosol particles, subrange 1a is an additional feature to describe the nucleation mode. Otherwise, Aitken- and accumulation-mode size ranges are sufficient to characterise cloud phenomena.

The aerosol, cloud and ice crystal size distributions are discretised into the bins according to the dry aerosol diameter, whereas the rain droplet size distribution is defined by the wet diameter of the droplet. Identical 2a and 2b size bins are used for aerosol, cloud droplets and ice. Such parallel bins are useful for tracking aerosol development through cloud activation, freezing and sublimation. Prognostic variables for

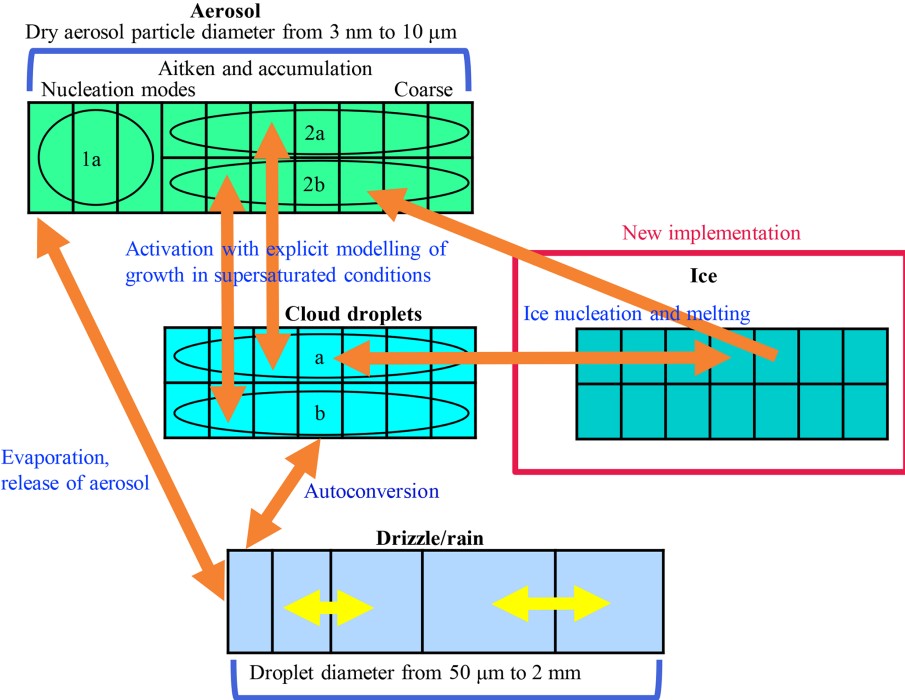

**Figure 1.** Bin scheme of UCLALES–SALSA with newly implemented particles; see also Fig. 1 in Tonttila et al. (2017).

each bin include aerosol number and the masses of all compounds (water, sulfate, dust, organic carbon, sea salt, nitrate and ammonium).

In UCLALES–SALSA, recently implemented processes involving ice crystals are droplet freezing, deposition of water vapour, sublimation, melting when $T > 0\,°C$, coagulation between different hydrometeors, sedimentation, and interactions with radiation (see also Fig. 1). Most of these processes are included in a similar way as in the previously published version of UCLALES–SALSA (Tonttila et al., 2017).

Regarding the scope of this study, we describe droplet freezing in higher detail. There are five mechanisms for droplet freezing, and they are all currently implemented in UCLALES–SALSA.

– Immersion freezing is possible for aqueous droplets that have an insoluble core, which in UCLALES–SALSA is either dust (DU) or black carbon (BC). The rate of heterogeneous germ formation in a supercooled droplet of water or solution is calculated mostly following Khvorostyanov and Curry (2000), and additional parameters are from Jeffery and Austin (1997), Khvorostyanov and Sassen (1998), Khvorostyanov and Curry (2004), and Li et al. (2013). See also Appendix A.

– Homogeneous freezing is possible for any aqueous droplet with or without insoluble particles. This is applied to the model according to Khvorostyanov and Sassen (1998). See also Appendix B.

– Deposition freezing is possible for dry insoluble aerosol at subsaturated conditions (RH < 100 %). This is implemented following Khvorostyanov and Curry (2000) and additional parameters from Hoose et al. (2010). See also Appendix C.

– Contact freezing is implemented in UCLALES–SALSA following Hoose et al. (2010) so that first the coagulation code is used to calculate collision rates between dry particles and liquid droplets, and then the immersion freezing code gives the freezing probability.

– Condensation freezing is implemented as a part of immersion freezing because these droplets can freeze during the modelled condensational growth.

In our simulations (Sect. 3.3), only immersion freezing is active. This is due to high temperatures, when homogeneous freezing is not possible, when the mixing state of INPs leads to aqueous droplets, and when deposition and contact freezing are not feasible.

Deposition of water, i.e. diffusion-limited condensation or evaporation of water vapour, is defined for aerosol when relative humidity (RH) is over 98 % (equilibrium assumed otherwise) and always for other hydrometeors. This is based on the analytical predictor of condensation (APC) scheme by Jacobson (2005) and implemented following Tonttila et al. (2017) (Eqs. 7 and 8). According to this definition, the particles compete for the available water vapour. For solids, the condensation equation does not require Kelvin or Raoult terms.

If ice sublimates, the immersed aerosol nuclei are added back to the aerosol population.

Activation of aerosols to cloud droplets happens when RH is over 100 % and aerosol wet diameter exceeds the critical limit corresponding to the resolved supersaturation. At this time, a certain proportion of activated aerosols (i.e. cloud condensation nuclei, CCN) is moved to cloud droplet bins.

Sedimentation is defined as before in Tonttila et al. (2017) and now extended for ice particles. For simulations in this study, a fall rate of ice particles is set as in Ovchinnikov et al. (2014).

Coagulation is implemented the same way as before and now also including ice particles. Coagulation is affected by diffusion, especially aerosols, and by sedimentation, especially large particles. In a collision, bigger particles absorb smaller particles.

Interaction with radiation is implemented either with the same four-stream radiative transfer solver (Fu and Liou, 1993) as in Tonttila et al. (2017) with an extension to include ice particles or parameterised as in Ovchinnikov et al. (2014). We used the latter method in our simulation. In the parameterised radiation, the net upward long-wave radiative flux is computed as a function of the liquid water mixing ratio profile. The effect of interaction with radiation can be seen in simulations: radiative cooling weakens after liquid water path decreases below a specific value.

Furthermore, UCLALES–SALSA was upgraded with minor bug fixes and improvements. For example, hygroscopicity is now calculated with $\kappa$-Köhler (Petters et al., 2006; Petters and Kreidenweis, 2007) instead of the previously used ZSR method (Zdanovskii–Stokes–Robinson method; Zdanovskii, 1936; Stokes and Robinson, 1966).

## 3   Results

### 3.1   Model evaluation

The model performance is evaluated by simulating a well-documented mixed-phase cloud case from the Indirect and Semi-Direct Aerosol Campaign (ISDAC) Arctic study (McFarquhar et al., 2011). This observation case has been used before for comparisons to LES models (e.g. Savre and Ekman, 2015; Fu and Xue, 2017). Ovchinnikov et al. (2014) presented an intercomparison of 11 LES models for this same case, in which initial profiles were based on aircraft observations in the mixed layer (Flight F31) and idealisation of a sounding on 26 April 2008 at Barrow, AK. Nine of those models had bulk two-moment microphysics and two of them bin microphysics.

We implemented in UCLALES–SALSA model runs with the same semi-idealised simulation setup given in Ovchinnikov et al. (2014) that attempted to minimise intermodel differences by applying identical descriptions for the following processes: surface properties and fluxes (fluxes set to zero),

large-scale forcings, radiation, cloud droplet freezing and ice growth processes and sedimentation, and the nudging of horizontal winds, potential temperature and water content above the altitude of 1200 m. In the simulations ice processes were excluded during the first 2 h, i.e. the spin-up period, to allow the mixed-layer turbulence and the warm stratus cloud to develop. After the spin-up, cloud droplets are allowed to freeze until a specified target ice concentration is reached (Morrison et al., 2011b). Ice shape is described with a mass-diameter parameterisation so that ice can be considered spherical particles with low effective density ($\rho = 84.5\,\mathrm{kg\,m^{-3}}$). Ice fall speed is related to the maximum dimension, while capacitance, which is used in the condensation equation, is modified from that of a sphere to $C = D/\pi$. Radiation and sedimentation were parameterised similar to Ovchinnikov et al. (2014). For the sake of simplicity, coagulation was switched off as in Ovchinnikov et al. (2014). Warm rain formation was switched off, allowing for more straightforward model intercomparison. Also, the warm rain mass mixing ratios would have been small due to a relatively small cloud droplet size in the simulated case. Aerosol size distribution is given as a sum of lognormal accumulation and coarse modes with concentrations of $159 \times 10^6$ and $6.5 \times 10^6\,\mathrm{kg^{-1}}$, mode mean diameters of 0.2 and 0.7 μm, and geometric standard deviations of 1.5 and 2.45, respectively. Aerosol is composed of ammonium bisulfate. During the simulations this aerosol size distribution provides on average $129 \times 10^6\,\mathrm{kg^{-1}}$ cloud droplets. These aerosol distribution parameters provide the best fit to the measured distributions below the liquid cloud layer (Earle et al., 2011; Ovchinnikov et al., 2014).

We ran UCLALES–SALSA for the three different simulation setups investigated in the Ovchinnikov et al. (2014) study: no ice (ICE0), average ice (ICE1) and high ice (ICE4) number concentration. The ice number concentration is the only variable that was changed between the evaluation simulations (0, 1 or $4\,\mathrm{L^{-1}}$). Liquid and ice water paths (marked LWP and IWP from now on), i.e. column-integrated mass values averaged over the horizontal domain, in these three cases show how water is distributed between ice and liquid phases depending on the ice crystal concentration.

Figure 2 compares the three UCLALES–SALSA simulations to the results presented in the Ovchinnikov et al. (2014) intercomparison paper. In the figure, LES model results from Ovchinnikov et al. (2014) are separated between bulk and bin microphysics to highlight the differences between microphysics schemes. First, Fig. 2a shows LWP for the baseline simulation without any ice (ICE0). It is evident that our model agrees well with the other 11 models. The simulated LWP of UCLALES–SALSA is in the middle of the model spread. Differences are most likely explained by different dynamical cores, which is also stated in Ovchinnikov et al. (2014). A more thorough testing of warm-phase cloud microphysics in UCLALES–SALSA was done in the Tonttila et al. (2017), and for the remainder of this work we will con-

centrate on examining the properties of the ice microphysics implementation.

Second, Fig. 2b and c present the LWP and IWP time series when the target ice number concentration is $1\,L^{-1}$, marked with ICE1. Again, LWP in UCLALES–SALSA matches the other models well, being at the lower end of the intermodel spread. As expected, the LWP growth rate is lower than in the ICE0 simulation, as some of the water vapour condenses onto ice crystals. Furthermore, IWP matches well, especially with other bin models in the Ovchinnikov et al. (2014) study.

Third, Fig. 2d depicts LWP time series with an ice number concentration of $4\,L^{-1}$, which can be regarded as high ice concentration and is marked with ICE4. After spin-up, LWP has a decreasing trend since the ice number concentration is so high that it consumes much of the water vapour. Subsequently, IWP in Fig. 2e increases rapidly after the spin-up and in UCLALES–SALSA reaches its peak value of $15.7\,g\,m^{-2}$ just before 4 h of simulation. It then decreases to a value of $9.4\,g\,m^{-2}$ at the end of the simulation. The reduction of IWP is caused by ice crystal precipitation at the surface and evaporation below the cloud.

Compared to the model results in Ovchinnikov et al. (2014), IWP in UCLALES–SALSA declines faster after the peak IWP has been reached in ICE4. One reason for this is that dry particle size is tracked in UCLALES–SALSA, and this seems to have an important effect on ice crystal sedimentation. Namely, sedimentation velocities and particle mixing (flux divergency) are calculated here for the dry size bins rather than bins tracking ice particle size. This reduces particle flux, especially in the lowest 200 m, leaving more particles there to evaporate. Evaporative cooling leads to a surface inversion which prevents the mixing of moist surface air. As such, the higher sensitivity to INP concentration is partly related to the initial conditions of the ISDAC case study. The other reason is related to the model-dependent technical details. Our test simulations (not shown) indicate that changing model options, such as flux limiter method, impact IWP and LWP so that the gap between UCLALES–SALSA and the other models decreases. In Ovchinnikov et al. (2014) it was also stated that when the ice number concentration gets higher the differences between models are more caused by discrepancies in microphysics than in cloud dynamics. This underlines the sensitivity and significance of microphysics.

To conclude, the spread between models, especially between bin and bulk microphysics models, gets wider as the prescribed ice number concentration gets larger and closer to the limit when the cloud glaciates completely. In UCLALES–SALSA this limit of full glaciation is lower than in other models in Ovchinnikov et al. (2014). This limit is further examined in Sect. 3.2.

## 3.2 Sensitivity on ice concentration

Motivated by simulated differences with the $4\,L^{-1}$ ice concentration, we wanted to further investigate how sensitive cloud glaciation is to changes in ice number concentration. In addition to ICE1 and ICE4 simulations, we performed simulations in which the target ice number concentration was 2, 3, 5 or $6\,L^{-1}$ (marked with ICE2, ICE3, ICE5 and ICE6, respectively). Figure 3 depicts the LWP and IWP evolution in all six UCLALES–SALSA simulations. The simulation time was extended to 24 h in those cases in which cloud still exists after 8 h (marked with a vertical line in Fig. 3). The simulation time was not extended any further because we do not see any major changes or trends in the last simulation hours.

Figure 3 shows that when the ice number concentration is set to a higher value, LWP decreases faster and cloud glaciates sooner. In simulations ICE4, ICE5 and ICE6, the cloud dissipates totally after glaciation. The cloud glaciation happens because water vapour condenses on the ice crystals at the expense of the cloud droplets. In simulations ICE1, ICE2 and ICE3, IWP stabilises to values of approximately 6.5, 10 and $12\,g\,m^{-2}$, respectively, towards the end of the simulation.

From Fig. 3 we can also see that LWP still increases during the first 8 h with ICE2 but not anymore with ICE3. With ICE1, the water paths of the cloud are very stable after 8 h of simulation. LWP decreases about $2\,g\,m^{-2}$, reaching a value of $44\,g\,m^{-2}$ at the end of the simulation. IWP is around $7\,g\,m^{-2}$ at the end of the simulation. LWP values for ICE2 and ICE3 simulations are around 22 and $18\,g\,m^{-2}$, and IWP values are 10 and $12\,g\,m^{-2}$ at the end of the simulation, respectively. These are close to ICE4 simulations presented in Ovchinnikov et al. (2014), and this illustrates the fine balance between co-existing liquid and ice phases.

These results show how sensitive the mixed-phase cloud is to ice number concentration either by showing how fast the cloud glaciates or when balance is reached. However, these are highly simplified due to the lack of real aerosol-dependent freezing and related feedback processes. These results also show the need for more detailed feedbacks since a constant ice number concentration is not a realistic assumption for real clouds.

## 3.3 Prognostic ice simulation

One of the unique features of our model is its ability to keep track of the chemical composition along with a sectional aerosol size distribution in the cloud phase. This allows us to model freezing processes related to an ice-nucleating compound like dust. Furthermore, parallel bins allow for analysing the relative contribution of e.g. dust particles (INPs) on ice formation. We call this prognostic ice because here freezing probability is related to dust aerosol, the mass and number concentrations of which are prognostic variables. We allow interactions between all hydrome-

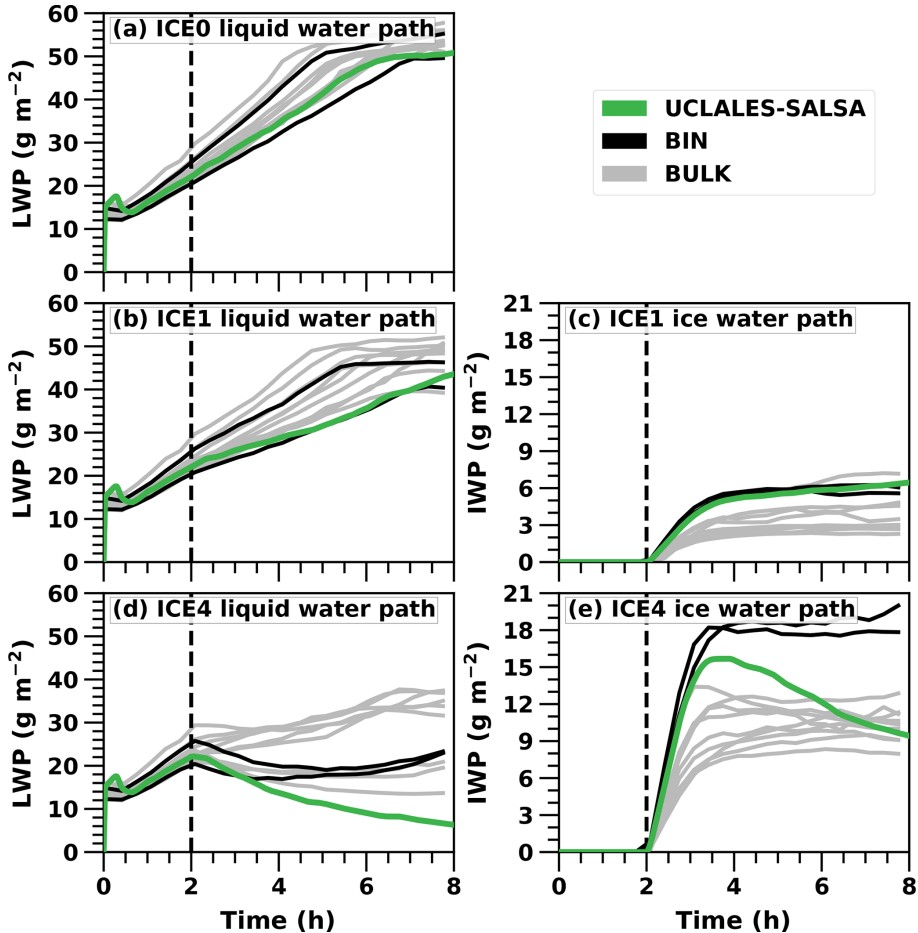

**Figure 2.** Liquid and ice water path time series in UCLALES–SALSA simulations with fixed ice number concentrations of 0, 1 and $4\,L^{-1}$ (ICE0, ICE1 and ICE4, respectively). Black and grey lines show results in the Ovchinnikov et al. (2014) study.

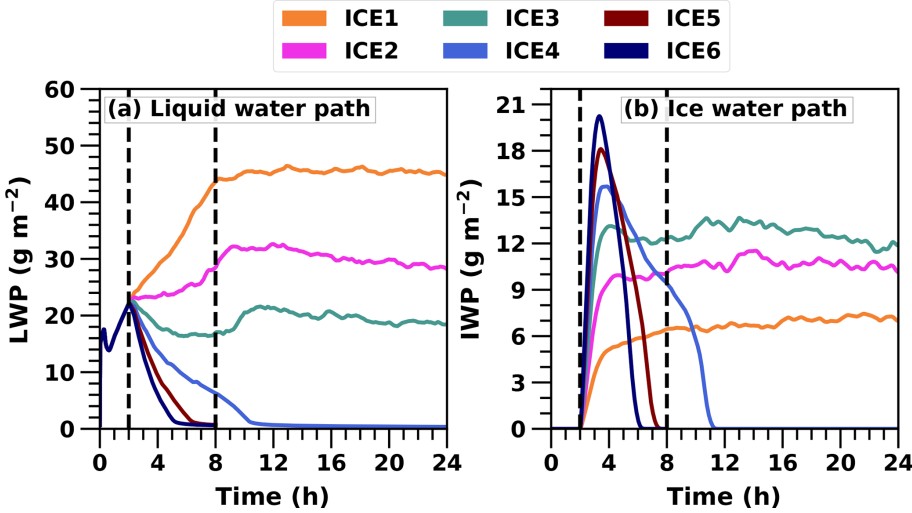

**Figure 3.** Liquid and ice water path time series in UCLALES–SALSA simulations with fixed ice number concentrations of 1, 2, 3, 4, 5 and $6\,L^{-1}$ (ICE1, ICE2, ICE3, ICE4, ICE5 and ICE6, respectively).

ors, and ice formation is modelled using the implemented ice nucleation theories, which relate ambient conditions and droplet properties to their freezing rates.

To see the difference between fixed and prognostic droplet freezing, we made a prognostic ice simulation that was targeted to have similar IWP during the first 8 h as in the simulation with ice number concentration of $4\,L^{-1}$ (ICE4) (see Sect. 3.1 and 3.2). This ICE4 simulation was selected for comparison because it is close to the tipping point at which cloud either stabilises or glaciates (see Sect. 3.2).

To achieve the target IWP, we adjusted the freezing properties accordingly of aerosols that can act as an INPs. The total number concentration and size distribution of the aerosol remain the same as in the fixed ice number simulations (Sect. 3.1 and 3.2); thus, they are the same as in Ovchinnikov et al. (2014). In the absence of more detailed aerosol observations, INP number concentration and mixing state as well as contact angle were considered adjustable parameters impacting ice nucleation ability. Here, contact angle represents the angle between the ice embryo and the ice nucleus in an aqueous medium.

First, in order to set the INP number concentration, we incorporated b bins (for more information about bin description, see Sect. 2 and Fig. 1). The proportion $x = 150 \times 10^{-6}$ of the total aerosol number concentration was partitioned in b bins as INPs. The proportion $(1-x)$ remained in the a bins. The resulting number concentrations of INPs in accumulation and coarse modes were $2385 \times 10^3$ and $975 \times 10^3\,kg^{-1}$, respectively.

Second, the INP mixing state was adjusted so that the particles in the b bins were set to have an insoluble dust core, 50 % of the dry mass and ammonium bisulfate for the other half. Here, dust acts as INPs.

Third, the freezing rate was adjusted by setting the cosine of the contact angle of dust to $m_{is} = 0.57$ (Eq. A3 in Appendix A).

It should be noted that the target IWP could have been reached using different combinations of INP mixing state, $x$ and $m_{is}$, but these simulations showed that the results depend mostly on the resulting ice number concentration rather than the applied parameterisation. These characteristics of aerosol are uniform throughout the whole simulation domain.

The simulation time for the prognostic ice run was set to 32 h. The water paths of the mixed-phase cloud are quite stable after that. The simulation time of ICE4, used to compare with the prognostic run, was not extended any further from 24 h since the cloud dissipates around 12 h of simulation.

As in the ICE4 simulation, in the prognostic ice simulation, droplet freezing was set to start after a spin-up of 2 h. Figure 4a and b illustrate that the prognostic ice and ICE4 simulations have similar IWP and LWP during the first 8 h. Hence, the targeting is successful and the initial conditions of the simulations match each other. After that, the prognostic ice simulation diverges from the ICE4 simulation.

Figure 4a shows that in the prognostic ice simulation LWP starts to increase after 4.5 h of simulation. This is caused by a decrease in ice number concentration (Fig. 4c) to such a low level, which allows more water vapour for condensation to liquid droplets. The same figure also depicts how the ice number concentration is set to a target value (simulation ICE4) and how the concentration is stable until the cloud dissipates. Figure 4d depicts how droplet number concentration lowers, especially right after the spin-up period when ice number concentration is increasing. However, changes in droplet number concentration are not the driving force behind complete removal of liquid phase. Figure 4e illustrates how the whole cloud with prognostic droplet freezing descends, and as the cloud in the ICE4 simulation is affected by entrainment both below and above the cloud, the cloud gets thinner and dissipates.

At the beginning of the prognostic ice run, the domain mean of dust-containing aerosols is approximately $27\,L^{-1}$. After 32 h of simulation the same mean value is about $13\,L^{-1}$. Here, the loss of INPs limits the ice number concentration. The mixed-phase cloud persists because the ice number concentration can change. This is so-called self-adjustment of INPs, which better reproduces the observed evolution of mixed-phase clouds since they are usually more resilient in observations than in models (Andronache, 2017; Morrison et al., 2011a). This is also in line with previous modelling studies, wherein prognostic INPs will reduce the number of ice crystals because of precipitation, thus allowing cloud liquid to be sustained (Fridlind et al., 2012; Solomon et al., 2015, 2018). The decrease in dust (INPs) mass concentration in different hydrometeor types is shown in Fig. 5. Dust is an efficient ice nucleus, so it will soon end up in ice crystals which are removed from the system by sedimentation (Fig. 5c). The free troposphere is the only source for the boundary layer dust, and the relevant mechanisms are entrainment and large-scale subsidence. Subsidence is described with a downward vertical velocity moving mass and energy. Entrainment in this case describes any other kind of mass exchange between cloud top and the free troposphere. For instance, subsidence is $0.004\,m\,s^{-1}$ at the cloud top and the aerosol number concentration in dust-containing b bins above the cloud is about TS1 $24\,000\,m^{-3}$, so the dust aerosol flux from the free troposphere is approximately $100\,m^{-2}\,s^{-1}$. Because radiative cooling is strengthening the supersaturation at the cloud top, the most CCN-active part of these entrained dust-containing particles can be activated immediately as cloud droplets. This can be seen as a higher dust mass concentration within cloud droplets in the upper layer of the cloud (Fig. 5b). If the temperature is low enough, these dust-containing droplets will subsequently freeze during the following time steps and therefore take part in preserving the mixed-phase cloud.

A more detailed examination of droplet activation and ice formation can be done by studying the time evolution of the size distribution. Figure 6 shows how particles of dif-

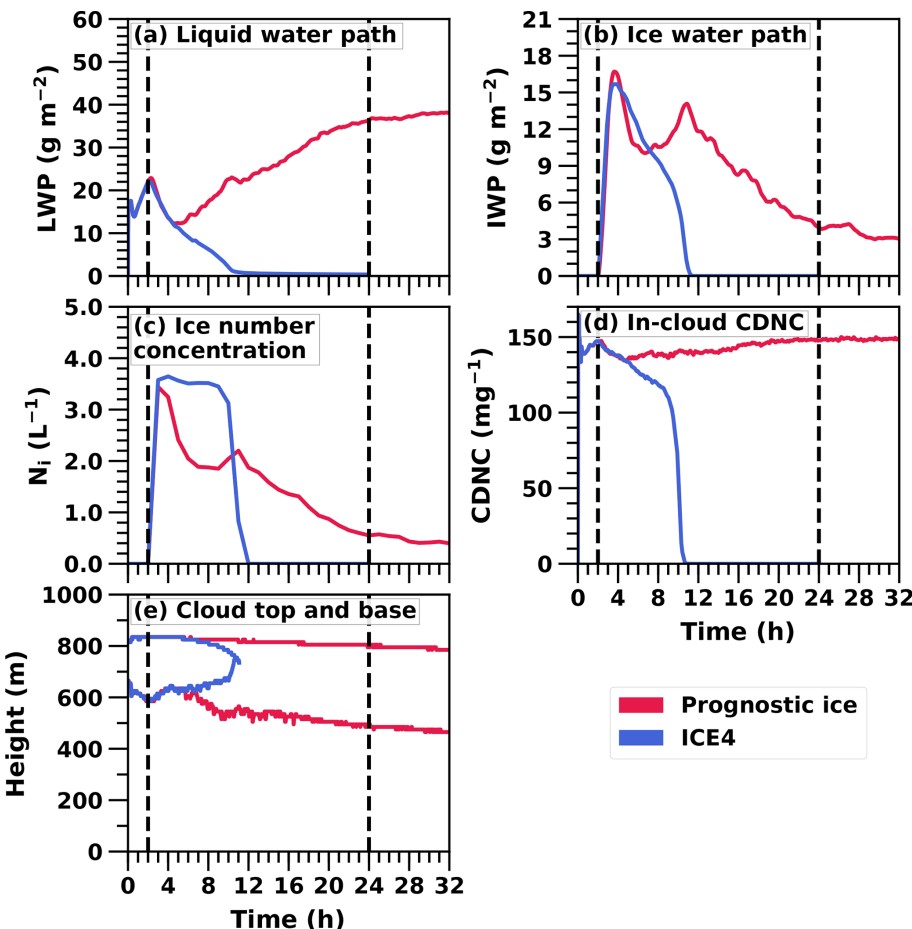

**Figure 4.** Time series of water paths, mean ice number concentration in icy regions, in-cloud cloud droplet number concentration (CDNC), and the cloud top and base of the 32 h UCLALES–SALSA simulation with prognostic droplet freezing ("Prognostic ice") compared with the 24 h UCLALES–SALSA simulation with a fixed ice number concentration of $4\,\mathrm{L}^{-1}$ (ICE4).

ferent sizes are partitioned between different hydrometeor types within the cloud layer. Figure 6c and d show how the larger particles freeze first and their number concentration decreases quickly as these particles deposit at the surface within falling ice hydrometeors and are removed from the system. Even though the entrainment from above is providing more particles, this is not fast enough to maintain the original concentration. Removal of the smaller INPs is slower, as those are less likely to activate as cloud droplets, and the resulting droplets are also less likely to freeze due to the smaller dust core area. However, with time and because of continuous mixing of the boundary layer, the smaller INPs are also eventually able to form cloud droplets within the strongest updrafts, and formed droplets will freeze within the cloud. This will lead to stabilisation of the aerosol size distribution. The increase in the total number of particles in bin 1 is a numerical artefact caused by the bin adjustment routine, which can move particles from one bin to another in order to keep the dry size within the predefined bin limits. When a large proportion of particles in bin 2 are activated as cloud

droplets, some of the remaining particles are moved to the smaller bin to avoid numerical problems. However, this numerical artefact does not affect the results.

Figure 7a and b illustrate the fact that supercooled liquid droplets are dominant in the upper layers of the mixed-phase cloud compared to ice crystals. Here the total ice number concentration stabilises at approximately $0.44\,\mathrm{L}^{-1}$, whereas it is obvious from Sect. 3.2 that a much higher concentration is needed to completely glaciate the cloud. Correspondingly, the cloud droplet number concentration stabilises at approximately $175\,\mathrm{cm}^{-3}$. Remarkably, these values are in line with aircraft observations (Flight F31) of this ISDAC case. The observed ice and cloud droplet number concentrations are $0.35\,\mathrm{L}^{-1}$ and $185\,\mathrm{cm}^{-3}$, respectively (McFarquhar et al., 2011; Savre and Ekman, 2015). The ice number concentration is also approximately 2 orders of magnitude less than the number concentration of efficient INPs above the cloud layer. From that we can estimate that an above-cloud INP concentration of the order of 0.1 to $1.0\,\mathrm{cm}^{-3}$ is enough to glaciate the cloud.

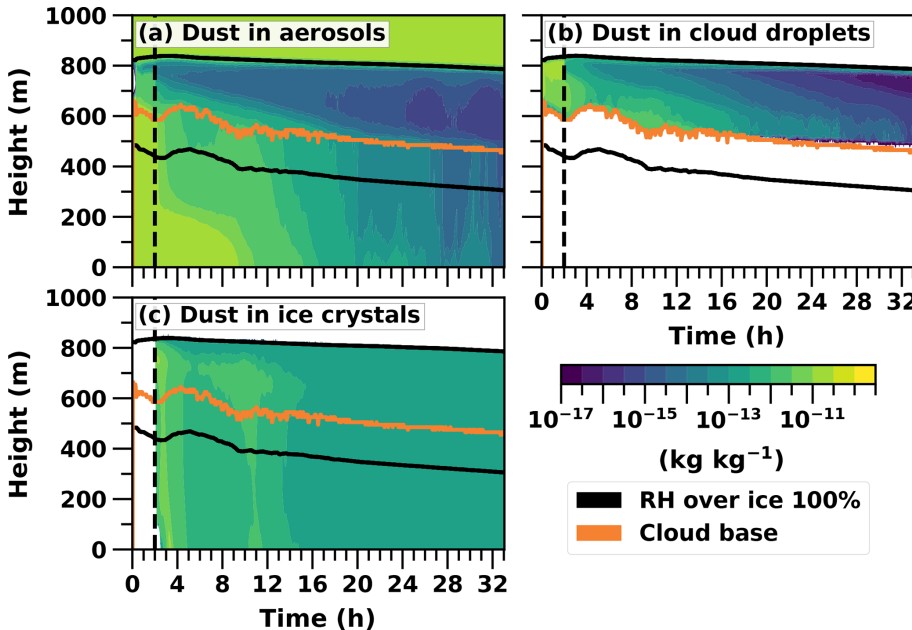

**Figure 5.** Logarithmic total mass mixing ratios ($\mathrm{kg\,kg^{-1}}$) of dust in different hydrometeors given as cloud profile time series in the UCLALES–SALSA simulation from the prognostic ice simulation. Cloud top is not plotted to keep the figure clearer because it is practically the same as the upper line of RH over ice.

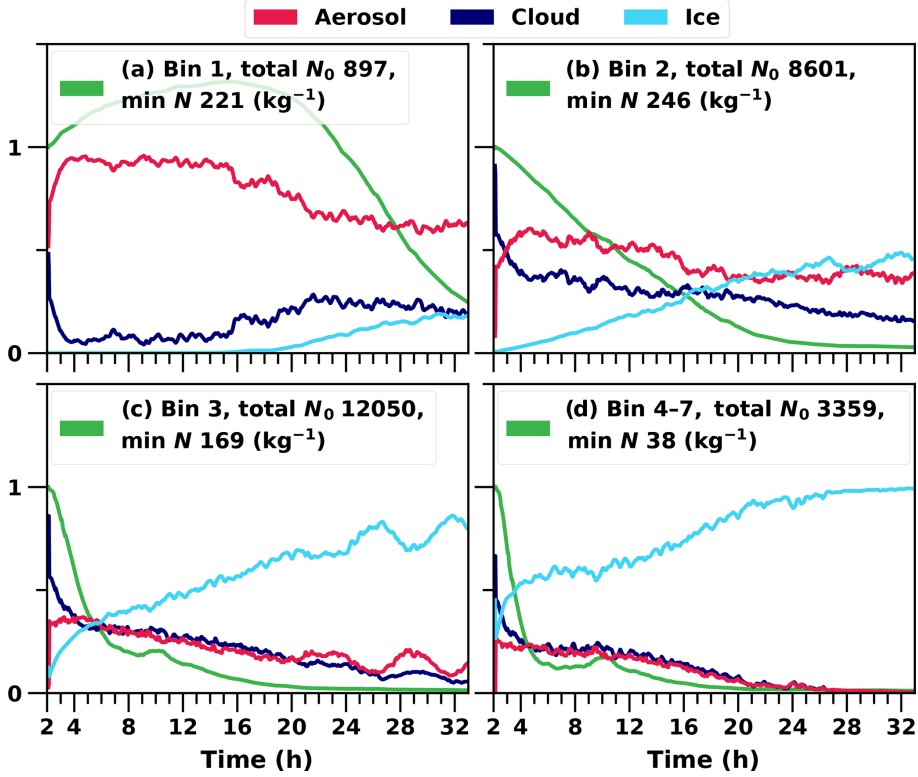

**Figure 6.** Relative proportions of hydrometeors at each time step in the cloud layer. A cloud layer is defined when both the cloud liquid water and ice mixing ratios are over 0.001 ($\mathrm{g\,kg^{-1}}$). The green line represents the relative change in the total number concentration in each bin.

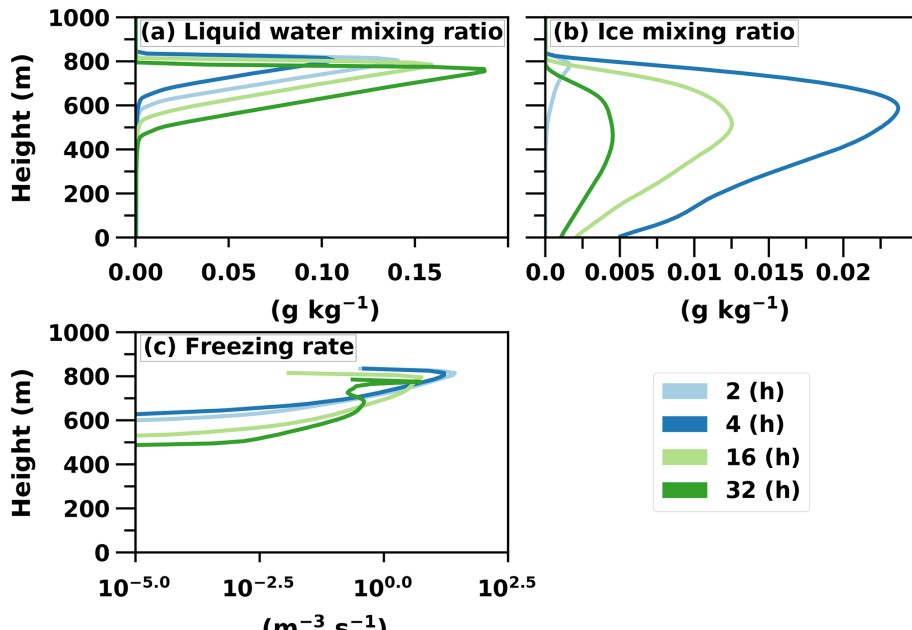

**Figure 7.** Vertical profiles of liquid water, ice and the freezing rate of droplets (nucleation rate) in the UCLALES–SALSA simulation with prognostic droplet freezing.

Figure 7c further illustrates an interesting behaviour of ice particle formation. At the beginning of the simulation ice particles are formed throughout the cloud, but later the most intensive formation takes place at the top of the cloud where fresh INPs are entrained into the cloud layer. However, the maximum supersaturation in these entraining downdrafts is so low that only the largest particles are able to form cloud droplets and consequently freeze. The smaller ones penetrate through the cloud layer as interstitial aerosol particles (i.e. unactivated particle) and are able to form cloud droplets (i.e. activate) and ice particles at the cloud base when they are recirculated back to the cloud with higher supersaturation. This can be seen in Fig. 8. Figure 8a shows that in size bin 2 cloud droplets and ice particles are more frequent in updrafts compared to Fig. 8b, which illustrates that aerosols are more favourable in downdrafts. Additionally, ice particles dominate in bigger sizes as aerosols freeze both in downdrafts and updrafts (size bin 3 shown in Fig. 8c and d). Simulated freezing in different vertical velocity conditions in other size bins does not differ from results shown already in Fig. 6. The lower peak at the end of the simulation in the vertical profile of freezing rate in Fig. 7c also shows how recirculated aerosols are frozen in the cloud layer. Such a phenomenon has been modelled before e.g. in Solomon et al. (2015); however, here the cloud is simulated with explicit calculation of in-cloud supersaturation and representation of aerosol size distribution and chemical composition. If activation is not modelled with this level of detail, activation and freezing might happen too early or late and in a wrong part of the cloud. Overall, Figs. 6, 7c and 8 nicely demonstrate how the

relative proportions of particles in different hydrometeors are size-dependent and how a sectional description for aerosols is required to be able to simulate such processes in LES models.

## 4 Conclusions

In this study we have extended our large-eddy model UCLALES–SALSA (Tonttila et al., 2017) for ice and mixed-phase clouds. The model has an exceptionally detailed sectional aerosol description for both aerosol number and chemical composition, which makes this model suitable for examining aerosol–cloud interactions and dynamics. Specifically, this allows for the description of an ice-nucleation-active material such as mineral dust, which can be used in calculating ice formation rates from the nucleation theory.

As the first step, we compared our model predictions with those from a mixed-phase cloud model intercomparison study (Ovchinnikov et al., 2014) to confirm the accuracy of the newly implemented ice microphysics. In this simplified model intercomparison setup, wherein any cloud droplet will freeze until a specified ice number concentration (from zero up to 4 particles $L^{-1}$) is reached, the focus is on cloud dynamics. In agreement with Ovchinnikov et al. (2014) and several other studies (e.g. Klein et al., 2009; Morrison et al., 2011b; Stevens et al., 2018) we conclude that microphysical details such as the fact that dry particle size is tracked in UCLALES–SALSA, while most other sectional models track the ice particle size, have an impact on predictions. Such details become more important close to the tipping

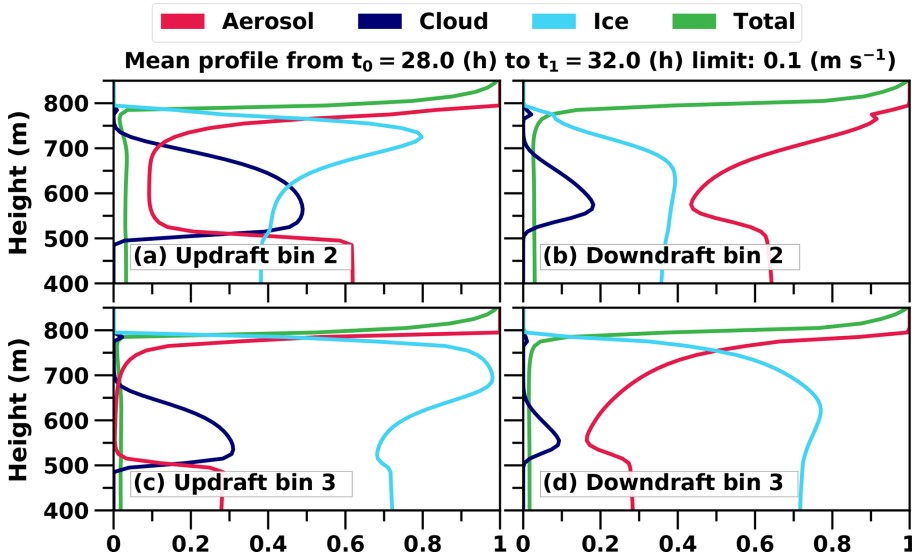

**Figure 8.** Mean profiles of the relative proportions of hydrometeors in different vertical velocity conditions. Velocities smaller than $0.1\,\mathrm{m\,s^{-1}}$ are neglected. Averaging is done from simulation hours 28 to 32. The green line shows the relative change in the total number of hydrometeors. Only bins 2 and 3 are shown since they provide additional information in comparison to Fig. 6.

point at which the further addition of ice particles leads to the rapid glaciation of the cloud.

In the second part, we constructed a case in which ice formation is modelled using a heterogeneous ice nucleation scheme and a prognostic ice-nucleating particle population containing mineral dust. This so-called prognostic ice simulation was designed so that it matched the previous fixed ice number concentration simulation in which the cloud was close to the tipping point. When the simulation with a fixed ice concentration showed complete glaciation after about 12 h, the prognostic ice simulation reached an equilibrium state which lasted up to end of the 32 h simulation. With this the prognostic simulation showed the importance of the self-adjustment of ice-nucleation-active particles. This is in good agreement with previous modelling studies (Fridlind et al., 2012; Solomon et al., 2015, 2018) and a observational study in which resilient mixed-phase clouds are seen together with relatively high ice nuclei concentrations (Filioglou et al., 2019).

Further examination of the prognostic ice simulation revealed that the efficient INPs entrained from the free troposphere are able to maintain the mixed-phase cloud with an ice particle number concentration on average 0.1 %–0.2 % of the INP concentration above the cloud. These entrained particles do not immediately form ice particles in the cloud top. The detailed analysis of the model outputs reveals how particle size and supersaturation-dependent cloud activation eventually control the formation of ice through immersion freezing. Some entrained INPs penetrate through the cloud as interstitial particles, get mixed within boundary layer air and contribute ice formation later when recycled back to the cloud.

Thus, the entrainment process is maintaining INP concentration in the whole boundary layer.

This study emphasises the benefits of the detailed aerosol–cloud–ice module within an LES model. In fact, UCLALES–SALSA is one of the few cloud-scale models (Fridlind et al., 2012; Khain et al., 2004; Savre and Ekman, 2015; Fu and Xue, 2017) with which details about aerosol and cloud droplet chemical composition can be utilised with a particle-level theoretical understanding of ice nucleation. The model will be a useful tool for mixed-phase cloud research, which has started to attract more widespread interest.

## Appendix A: Immersion freezing

The rate $J$ of heterogeneous germ formation by immersion freezing is a function of temperature $T$ in Kelvin, the radius of insoluble substrate $r_N$ and the equilibrium saturation ratio $S_w$ at the droplet surface based on Köhler theory; it is determined as

$$J(T, r_N, S_w) = C_{\text{het}} \exp\left[-\frac{\Delta F_{\text{act}} + \Delta F_{\text{cr}}}{kT}\right](\text{s}^{-1}),$$

$$C_{\text{het}} = \frac{kT}{h} c_{1\,s} 4\pi r_N^2, \tag{A1}$$

where $k$ and $h$ are the Boltzmann and Planck constants, $\Delta F_{\text{act}}$ is the activation energy at the solution–ice interface, $\Delta F_{\text{cr}}$ is the critical energy of germ formation, $C_{\text{het}}$ is the normalising function, $r_N$ is the radius of an insoluble fraction of an aerosol particle (INP), and $c_{1\,s}$ is the concentration of water molecules adsorbed on $1\,\text{cm}^{-2}$ of a surface (Eq. 2.1 in Khvorostyanov and Curry, 2004). The parameter values used are $C = 1.7 \times 10.999850^{10}\,\text{N m}^{-2}$ and $c_{1\,s} = 1 \times 10^{19}\,\text{m}^{-2}$.

Activation energy $\Delta F_{\text{act}}$ is calculated based on Eq. (15) in Jeffery and Austin (1997):

$$\Delta F_{\text{act}} = RT\left(\frac{B}{T - T_*} - \ln\frac{D_*}{D_0}\right)/N_A, \tag{A2}$$

where $T$ is temperature in Kelvin, $R$ is the molar gas constant, $N_A$ is the Avogadro constant, and parameter values $B = 347$, $T_* = 177$, $D_* = 4.17$ and $D_0 = 349$ for $p = 1$ bar are gained from Table 2 in Khvorostyanov and Curry (2004).

Critical energy is based on Eq. (2.10) in Khvorostyanov and Curry (2000):

$$\Delta F_{\text{cr}} = \frac{4\pi}{3}\sigma_{\text{is}} r_{\text{cr}}^2 f(m_{\text{is}}, x) - \alpha r_N^2 (1 - m_{\text{is}}), \tag{A3}$$

where

$$\sigma_{\text{is}} = 28 \times 10^{-3} + 0.25 T_c \times 10^{-3}\,\text{J m}^{-2} \tag{A4}$$

is the surface tension between ice and solution and where $T_c$ is temperature in degrees Celsius (Khvorostyanov and Sassen, 1998). The ice germ radius is

$$r_{\text{cr}} = \frac{\sigma_{\text{is}}}{\rho_{\text{ice}} L_m^{\text{ef}} \ln\left(\frac{T_0}{T} S_W^G\right) - C\epsilon^2}, \tag{A5}$$

where $\rho_{\text{ice}} = 900\,\text{kg m}^{-3}$ is the density of ice, and $T_0 = 273.15\,\text{K}$.

$$L_m^{\text{ef}} = \left(79.7 + 0.708 T_c - 2.5 \times 10^{-3} \times T_c^2\right)$$
$$\times 4.1868 10^3\,\text{J kg}^{-1} \tag{A6}$$

is the effective latent heat of melting (Eq. 6 in Khvorostyanov and Sassen, 1998). The dimensionless parameter is

$$G = \frac{RT}{M_w L_m^{\text{ef}}}, \tag{A7}$$

where $M_w$ is the molar mass of water (Eq. 2.7 in Khvorostyanov and Curry, 2000).

The shape factor $f$ is defined as a function of the ratio $x = r_N/r_{\text{cr}}$ and $m = m_{\text{is}}$. It is gained from Eq. (2.9) in Khvorostyanov and Curry (2000), originally from Fletcher (1962).

$$2f(m, x) = 1 + \left[(1 - mx)/\phi\right]^3 + x^3\left(2 - 3\psi + \psi^3\right)$$
$$+ 3mx^3(\psi - 1), \psi = (x - m)/\phi, \phi$$
$$= \left(1 - 2mx + x^2\right)^{1/2} \tag{A8}$$

The case-dependent parameters $\epsilon$, which is the elastic strain produced in an ice embryo by the insoluble substrate, $\alpha$, which is the relative area of active sites, and $m_{\text{is}}$, which is the cosine of the contact angle, are defined in our results (Sect. 3.3) to be

$$\epsilon = 0$$
$$\alpha = 0 \tag{A9}$$
$$m_{\text{is}} = 0.57.$$

The $m_{\text{is}}$ was used as a targeting parameter since the simulation tests were found to be very sensitive for this parameter. Other case-dependent parameters $\epsilon$ and $\alpha$ were not altered and had their default values.

## Appendix B: Homogeneous freezing

Homogeneous freezing is possible for any aqueous droplet with or without insoluble particles. This is applied to the model according to Khvorostyanov and Sassen (1998).

The number of crystals formed by homogeneous nucleation due to the freezing of supercooled pure water or deliquescent condensation nuclei is described by Eq. (1) in Khvorostyanov and Sassen (1998):

$$J = 2N_c\left(\frac{\rho_w kT}{\rho_{\text{ice}} h}\right)\sqrt{\frac{\sigma_{\text{is}}}{kT}}\exp\left[-\frac{\Delta F_{\text{act}} + \Delta F_{\text{cr}}}{kT}\right], \tag{B1}$$

where $k$ and $h$ are the Boltzmann and Planck constants, $\rho_w$ is the density of water, $\rho_{\text{ice}}$ is the density of ice (same as in A) and $N_c = 5.85 \times 10^{16}\,\text{m}^{-2}$ is the number of water molecules contacting a unit area of ice germ (Khvorostyanov and Curry, 2000).

Case-dependent activation energy $\Delta F_{\text{act}}(T) = 0.694 \times 10^{-19} \times (1 + 0.027(T_c + 30))\,\text{kg m}^2\,\text{s}^{-2}$ when $T_c < -30\,°\text{C}$ (Khvorostyanov and Sassen, 1998).

The effective latent heat of melting $L_m^{\text{ef}}$ is the same as in Eq. (A6). The dimensionless parameter $G$ is the same as in Eq. (A7). The surface tension between ice and solution $\sigma_{\text{is}}$ is the same as in Eq. (A4).

The ice germ radius is defined as

$$r_{cr} = \frac{\sigma_{is}}{\rho_{ice} L_m^{ef} \ln\left(\frac{T_0}{T} S_W^G\right)}, \tag{B2}$$

which is the same as in Eq. (A5) with $\epsilon = 0$ (Khvorostyanov and Curry, 2000).

Hence, we get the critical energy of germ formation (Khvorostyanov and Sassen, 1998, Eq. 9b):

$$\Delta F_{cr} = \frac{4}{3}\pi\sigma_{is} r_{cr}^2. \tag{B3}$$

## Appendix C: Deposition freezing

Deposition freezing is possible for dry insoluble aerosol at subsaturated conditions (RH < 100 %). This is implemented following Khvorostyanov and Curry (2000) and additional parameters from Hoose et al. (2010).

The rate of germ formation $J$ (s$^{-1}$) through deposition freezing is defined as in Khvorostyanov and Curry (2000, Eq 2.13) The pre-exponential factor (kinetic coefficient) is about

$$J = 10^{30} \times r_n^2 \exp\left[-\frac{\Delta F_{act} + \Delta F_{cr}}{kT}\right], \tag{C1}$$

where $r_n$ is the radius of the insoluble portion of the droplet, $k$ is Boltzmann's constant and $T$ is temperature. The pre-exponential factor (kinetic coefficient) is $10^{26}$ (cm$^{-2}$)$r_n^2$ (Fletcher, 1962). Here, the case-dependent activation energy $\Delta F_{act}$ is set to zero (Khvorostyanov and Curry, 2000).

Surface tension between ice and vapour (Hoose et al., 2010) is calculated with

$$\sigma_{iv} = \left[(76.1 - 0.155T_c) + (28.5 + 0.25T_c)\right]$$
$$\times 10^{-3}\,\mathrm{J\,m^{-2}}. \tag{C2}$$

The ice germ radius $r_{cr}$ (Khvorostyanov and Curry, 2000, Eq. 2.12) is defined as

$$r_{cr} = \frac{2\sigma_{iv}}{\left(R_v \frac{\rho_{ice}}{M_w} T \ln S_i - C\epsilon^2\right)}, \tag{C3}$$

where $S_i$ is the water vapour saturation ratio over ice, $T$ is temperature and $C$ is constant at $1.7 \times 10^{10}$ (N m$^{-2}$).

From previous values we get the critical energy of germ formation:

$$\Delta F_{cr} = \frac{4}{3}\pi\sigma_{iv} r_{cr}^2 f(m, x), \tag{C4}$$

where the shape factor $f(m, x)$ is defined as in Eq. (A8).

*Code and data availability.* The source code of the model is available from GitHub at https://github.com/UCLALES-SALSA/ UCLALES-SALSA (last access: 12 September 2019, Ahola et al., 2019) under release tag IceV1.0 and release name Ice microphysics V1.0. Model output data are available at http://urn. fi/urn:nbn:fi:att:5144df1e-4cdf-4d5a-af46-a545ebaa4460 (last access: 6 July 2020, Ahola et al., 2020). Figures are plotted with https: //github.com/JaakkoAhola/LES-ice-03plotting (last access: 10 August 2020, Ahola et al., 2020) under release tag v1.1.4.

*Author contributions.* JA made the simulations with the help of TR. JA analysed the data with help from TR and SR. JA wrote the paper with comments from HarK, HanK, TR, SR and JT. JA, TR, JT, HarK and SR have contributed to developing the UCLALES–SALSA model. HanK supervised the project.

*Competing interests.* The authors declare that there is no conflict of interest.

*Acknowledgements.* Mikhail Ovchinnikov is acknowledged for providing the simulation data of Ovchinnikov et al. (2014).

*Financial support.* This research has been supported by the European Research Council H2020 Research Infrastructures (ECLAIR (grant no. 646857)) and the Academy of Finland, Luonnontieteiden ja Tekniikan Tutkimuksen Toimikunta (grant no. 322532).

*Review statement.* This paper was edited by Toshihiko Takemura and reviewed by two anonymous referees.

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

**Remarks from the typesetter**

TS1 Please provide an explanation of why these values need to be changed. Changes in values have to be approved by the handling editor. Thanks.