# Peer review of "Modelling mixed-phase clouds with large-eddy model UCLALES-SALSA"

_Atmospheric Chemistry and Physics, 2019_

## Referee Comment (RC1) · Anonymous Referee #1 · 16 Feb 2020

Review for:

**"Modelling mixed-phase clouds with large-eddy model UCLALES-SALSA"**
*by J. Ahola et al.*

**General Comments:**

This paper offers a description of the microphysical updates regarding freezing processes in the LES model UCLALES-SALSA. A cloud case observed during ISDAC, that has been used for LES intercomparisons in the past, is also simulated here. This demonstrates the general agreement of the model with other LESs that are widely used for the study of mixed-phase clouds. A comparison of the newly implemented prognostic treatment of ice nucleation to a more simplified method is also presented. This paper will be useful to future users of UCLALES-SALSA, as it will serve as reference for the model's ice nucleation scheme. The few scientific findings are also interesting, specifically the role of INP treatment in cloud glaciation time and the impact of entrained INPs on ice formation throughout the cloud layer. For these reasons I recommend the paper for publication. However, I have a few suggestions that aim to (1) improve the documentation of the implemented freezing processes and (2) expand the scientific analysis and thus increase the scientific impact.

**Major Comments:**

(1) Since this paper will likely serve as a documentation of the freezing processes in future studies conducted with this model, I recommend to provide a description of all processes in the Appendix, not just the immersion mode.

(2) The prognostic simulation is conducted with assumed aerosol concentrations to reconstruct an IWP similar to the ICE4 experiment. However I recommend to use aerosol measurements from ISDAC in an additional simulation (e.g. as in Savre and Ekman 2015) and compare the results to the observations. If the prognostic scheme results in good agreement with reality or not is a critical piece of information for the cloud modelling community. Moreover, you can conduct a few more sensitivity simulations and activate other freezing processes as well, and show how these experiments compare with microphysical measurements.

**Minor Comments:**

**Line 40:** Do you mean that a high aerosol load is associated with higher occurrence of mixed-phase clouds or with more liquid in the mixed-phase clouds? Please clarify.

**Lines 246-249:** I don't see any point in comparing with observations since you simulated random aerosol conditions and not the observed.

Both INP and IN terms are used. I suggest to use the same term throughout the text for consistency (I think 'INP' has become more popular in the past few years)

---

## Referee Comment (RC2) · Anonymous Referee #2 · 7 Mar 2020

This study adds a heterogenous ice nucleation parameterization to the UCLALES-SALSA model. The model is tested with fixed ice crystal number concentration by using a case from the ISDAC campaign that was the focus of an intercomparison study. This paper is well written, and the figures clearly illustrate the main points. As to the results of the study, allowing prognostic INP will reduce the number of ice crystals because of precipitation, causing there to be more sustained cloud liquid, but how is this a new result? Many studies have already shown this (Fridlind et al. 2012; Solomon et al. 2015; Solomon et al. 2018). Also, the variability in the control studies differ significantly from the ISDAC intercomparison, which needs to be explained. Also, it needs to be explained how aerosol concentration above cloud top were chosen and what role the prognostic CCN is playing in the simulations. This model will be a very

useful tool for studying mixed-phase cloud processes, but I think this study is better suited for a technical report than a scientific publication.

Major comments:

1) Need to include basic detailed about the model in Section 2 even though they may be available in other papers. All details needed to understand the simulations need to be included in this section (CCN activation, etc).

2) This model is clearly more sensitive to ice formation than all the models included in the ISDAC intercomparison. It is important to understand why to understand the sensitivity studies with the new ice nucleation parameterization.

3) How is droplet number concentration specified in the ISDAC ICE4 simulation? Is this prognostic? If so, it would be insightful to see the droplet number concentration in Figure 4. Is this why the results are so different than the intercomparison?

4) It is not clear how the artificial movement of aerosols between bins for numerical stability is affecting the results (lines 240-243).

5) Please explain why droplet freezing occurs throughout the cloud while for the same case Savre and Ekman (2015) found droplet freezing at cloud top. A more detailed discussion of how aerosols and droplet and ice crystal activation are represented in the two models is needed to understand why simulations in the two studies differ.

6) Lines 275-277: More details of the simulations are needed to understand whether this is a correct statement.

Minor comments:

1) Line 198: "….concentration is was…". Please reword.

2) Line 202: "was adjusted"

3) Line 203: "represents"

---

## Author Comment (AC1) · 6 May 2020

**Response to the comments about the submitted paper: Modelling mixed-phase clouds with large-eddy model UCLALES-SALSA, Ref. ACP-2019-1182.**

Dear Editor, dear Reviewers, we would like to thank the Editorial Board for considering our paper for publication in ACP and the reviewers for their constructive comments. We have addressed all of them and modified the paper accordingly. Our detailed answers follow. Text from the original manuscript that has been removed in the revised manuscript is marked in red. New text in the revised manuscript is marked in blue.

**Answers to Reviewer 1**

**General comment R1.1** This paper offers a description of the microphysical updates regarding freezing processes in the LES model UCLALES-SALSA. A cloud case observed during ISDAC, that has been used for LES intercomparisons in the past, is also simulated here. This demonstrates the general agreement of the model with other LESs that are widely used for the study of mixed-phase clouds. A comparison of the newly implemented prognostic treatment of ice nucleation to a more simplified method is also presented. This paper will be useful to future users of UCLALES-SALSA, as it will serve as reference for the model's ice nucleation scheme. The few scientific findings are also interesting, specifically the role of INP treatment in cloud glaciation time and the impact of entrained INPs on ice formation throughout the cloud layer. For these reasons I recommend the paper for publication. However, I have a few suggestions that aim to (1) improve the documentation of the implemented freezing processes and (2) expand the scientific analysis and thus increase the scientific impact.
**Answer to R1.1** We thank the referee for these comments and do our best to improve the manuscript according to the suggestions.

**Major comment R1.2** Since this paper will likely serve as a documentation of the freezing processes in future studies conducted with this model, I recommend to provide a description of all processes in the Appendix, not just the immersion mode.
**Answer to R1.2** We have now provided a detailed description for homogeneous and deposition freezing processes in the Appendices.

**Major comment R1.3** The prognostic simulation is conducted with assumed aerosol concentrations to reconstruct an IWP similar to the ICE4 experiment. However I recommend to use aerosol measurements from ISDAC in an additional simulation (e.g. as in Savre and Ekman 2015) and compare the results to the observations. If the prognostic scheme results in good agreement with reality or not is a critical piece of information for the cloud modelling community. Moreover, you can conduct a few more sensitivity simulations and activate other freezing processes as well, and show how these experiments compare with microphysical measurements.
**Answer to R1.3** There are some aerosol measurements covering number size distributions and bulk chemical composition but information on ice nucleation activity of different compounds is missing. When this information is missing, predicting ice number concentration is uncertain. For this reason, in the prognostic simulation (Sect. 3.3), we selected dust as a common INP type and a reasonable mixing state, and adjusted the contact angle to yield a similar IWP to that in the ICE4 experiment in the beginning of the simulation. A similar approach was used by Savre and Ekman (2015). As described in the manuscript, this approach was chosen so that our simulated IWP could be compared with the corresponding ICE4 simulations and because then the ice number concentration is close to the tipping point where cloud either stabilises or glaciates. It also turned out that this initially adjusted IWP eventually lead to number concentration values and cloud persistence seen in measurements.

Although it is not mentioned in the manuscript, we made various quick sensitivity simulations and tested different freezing mechanisms. These sensitivity simulations examined, for example, INP mixing state and contact angle. For the temperatures in the ISDAC case, both immersion and contact freezing can produce ice and the relevant mechanism depends on the mixing state of the INP. Test simulations showed that the outcome depends mostly on the resulting ice number

concentration rather than the actual mechanism. For this reason, we showed just one prognostic case based on immersion freezing, which is the dominating mechanism for mixed-phase clouds. In answer to the question R1.5 we further clarify how our aerosol conditions match with observations.

Changes in the manuscript:

Remark about freezing processes in Sect. 2:

In our simulations (Sect. 3.3), only immersion freezing is active. This is  due to high temperatures, when homogeneous freezing is not possible, and mixing state of the INP leading to aqueous droplets, when deposition and contact freezing are not feasible.

3rd chapter of Sect. 3.3 rewritten as:

To achieve the target IWP, we adjusted accordingly the freezing properties of aerosols that can act as an INP. The total number concentration and  size distribution of the aerosol remain the same as in the fixed ice number simulations (Sect. 3.1 and 3.2), thus they are the thus same as in Ovchinnikov et al. (2014) . In the absence of more detailed aerosol observations, INP number concentration and mixing state, and contact angle were considered as adjustable parameters impacting ice nucleation ability. Here, contact angle represents the angle between the ice embryo and the ice nucleus in an aqueous medium.

First, in order to set the INP number concentration, we incorporated b bins (For bin description, see Sect. 2 and Fig. 1). Proportion $x = 0.015$ of the total aerosol number concentration was partitioned in b bins as INPs. Proportion $(1-x)$ of  remained in a bins. Resulting number concentrations of INPs in accumulation and coarse modes were $238.5 \times 10^3$ and $9.75 \times 10^3 kg^{-1}$, respectively.

Second, the INP mixing state was adjusted so that the particles in the b bins were set to have an insoluble dust core, 50% of the dry mass, and ammonium bisulphate for the other half. Here, dust acts as the INP.

Third, the freezing rate was adjusted by setting the cosine of the contact angle of dust to $m_{is} = 0.57$ (Eq. A3 in Appendix A).

It should be noted that the target IWP could have been reached using different combinations of INP mixing state, $x$ and $m_{is}$ but these simulations showed that the results depend mostly on the resulting ice number concentration rather than the applied parametrisation. These characteristics of aerosol are uniform throughout the whole simulation domain.

**Minor comment R1.4** Line 40: Do you mean that a high aerosol load is associated with higher occurrence of mixed- phase clouds or with more liquid in the mixed-phase clouds? Please clarify.
**Answer to R1.4** We clarified in the manuscript that a high aerosol load is associated with higher occurrence of mixed-phase clouds.

**Minor comment R1.5** Lines 246-249: I don't see any point in comparing with observations since you simulated random aerosol conditions and not the observed.

**Answer to R1.5** We have sharpened our statement how the initial aerosol number concentration and size distribution is the same as in Ovchinnikov et al (2014). Ovchinnikov et al (2014) cites that these parameters provide the best fit to the measured distributions below the liquid cloud layer (Earle et al., 2011). However, in our prognostic simulation, we altered the number of aerosols that contain an ice nucleating core and the contact angle between the ice embryo and the ice nucleus. This latter quality of the aerosol condition, i.e. freezing rate efficiency, is not available from observations.

Changes in the manuscript:

3rd chapter of Sect. 3.3 rewritten as already given in the answer to the question R1.3.

The paragraph that the referee mentioned:

Figures 7a and 7b illustrate  that super-cooled liquid droplets are dominant in the upper layers of the mixed-phase cloud compared to ice crystals. Here the total ice number concentration stabilises at approximately $0.44L^{-1}$, whereas it is obvious from Sect. 3.2 that a much higher concentration is needed to completely glaciate the cloud. Correspondingly, the cloud droplet number concentration  stabilises at approximately 175 $cm^{-3}$. Remarkably, these values are in  line with aircraft observations (Flight F31) of this ISDAC case. The observed ice and cloud droplet number concentration are $0.35L^{-1}$ and $185\ cm^{-3}$, respectively (McFarquhar et al., 2011; Savre and Ekman, 2015).  Ice number concentration is also approximately two orders of magnitude less than the number concentration of efficient  INPs above the cloud layer. From that we can estimate that the concentration of  INPs entraining from above the cloud should be in the order of 0.1 to $1.0\ cm^{-3}$ to glaciate the cloud.

**Minor comment R1.6** Both INP and IN terms are used. I suggest to use the same term throughout the text for consistency (I think 'INP' has become more popular in the past few years)

**Answer to R1.6** Manuscript was corrected to use only INP terms as suggested.

**Answers to Reviewer 2**

**General comment R2.1** This study adds a heterogenous ice nucleation parameterization to the UCLALES- SALSA model. The model is tested with fixed ice crystal number concentration by using a case from the ISDAC campaign that was the focus of an intercomparison study. This paper is well written, and the figures clearly illustrate the main points.

As to the results of the study, allowing prognostic INP will reduce the number of ice crystals because of precipitation, causing there to be more sustained cloud liquid, but how is this a new result? Many studies have already shown this (Fridlind et al. 2012; Solomon et al. 2015; Solomon et al. 2018).

Also, the variability in the control studies differ significantly from the ISDAC intercomparison, which needs to be explained.

Also, it needs to be explained how aerosol concentration above cloud top were chosen and what role the prognostic CCN is playing in the simulations.

This model will be a very useful tool for studying mixed-phase cloud processes, but I think this study is better suited for a technical report than a scientific publication.
**Answer to R2.1** We thank the referee for these comments and agree that the part describing the ice microphysics of UCLALES-SALSA for the first time is technical in nature. However, we argue that the manuscript also contains new scientific findings, such as the impact of entrained INPs on ice formation throughout the cloud layer, which Referee 1 also highlighted. We have added citations to the articles the referee mentioned and now state that our findings regarding prognostic INP and cloud resilience are in line with previous modelling studies. Here below, we answer in detail to the remarks raised in this general question.

We cite the articles the referee mentioned and state the study is also in line with previous modelling studies regarding prognostic INP and cloud resilience.

Changes in the manuscript (Sect. 3.3):

In the beginning of the prognostic ice run, domain mean of dust containing aerosols is approximately $27L^{-1}$. After 32 hours of simulation the same mean value is about $13L^{-1}$. Here, the loss of INPs limits the ice number concentration. The mixed-phase cloud persists because the ice number concentration can change. This is so-called self-adjustment of INPs which better reproduces observed evolution of mixed-phase clouds since usually they are more resilient in observations than in models(Andronache, 2017; Morrison et al., 2011a). This is also in line with previous modelling studies, where prognostic INP will reduce the number of ice crystals because of precipitation, thus allowing cloud liquid to sustain (Fridlind et al., 2012; Solomon et al., 2015, 2018).

Changes in Conclusions:

In the second part, we constructed a case where ice formation is modelled using a  heterogeneous ice nucleation scheme and a prognostic ice nucleating particle population containing mineral dust. This so-called prognostic ice simulation was designed so that it matched with

the previous fixed ice number concentration simulation where the cloud was close to the tipping point. When the simulation with fixed ice concentration showed a complete glaciation after about 12 hours, the prognostic ice simulation reached an equilibrium state which lasted up to end of the 32 hour simulation. With this the prognostic simulation showed the importance of the self-adjustment of ice nucleation active particles. This is in good agreement with previous modelling studies (Fridlind et al., 2012; Solomon et al., 2015, 2018) and a observational study where resilient mixed-phase clouds are seen together with relatively high ice nuclei concentrations (Filioglou et al., 2019).

We reply to the comment about the variability in the control studies in the answer to question R2.3.

We specify in the manuscript that the initial aerosol concentration and size distribution is uniform throughout the domain thus the concentration above cloud top is not any different than elsewhere. Answer to the comment about role of prognostic CCN is given in the answers to questions R2.2 and R2.4.

**Major comment R2.2** Need to include basic detailed about the model in Section 2 even though they may be available in other papers. All details needed to understand the simulations need to be included in this section (CCN activation, etc).
**Answer to R2.2** We rewrote the Model description (Sect. 2) particularly regarding radiative cooling, CCN activation and ice microphysics. A more detailed description of freezing processes is also given in the Appendices.

Changes in the Sect. 2 in the revised manuscript:

First paragraph of Sect. 2.

The UCLALES-SALSA (Tonttila et al. 2017) model consists of two components: first, the widely used large eddy simulator UCLALES (Stevens et al., 1999, 2005), and second the aerosol bin microphysics model SALSA (Sectional Aerosol module for Large-Scale Applications) (Kokkola et al., 2008; Tonttila et al., 2017; Kokkola et al., 2018). UCLALES handles e.g. surface fluxes, transportation of microphysical prognostic variables and atmospheric dynamics including turbulence. The previous version of UCLALES-SALSA incorporated interactions between aerosols, clouds and drizzle (Tonttila et al. 2017). Now we have extended the model with a description for ice crystals. In this study, we focus on how ice crystals and ice nucleating particles (INP) interact with clouds while tracking sectional aerosol size distribution.

5th paragraph of Sect. 2. and onwards rewritten:

[revised manuscript text omitted]

**Major comment R2.3** This model is clearly more sensitive to ice formation than all the models included in the ISDAC intercomparison. It is important to understand why to understand the sensitivity studies with the new ice nucleation parametrisation.

**Answer to R2.3** One contributing reason is that dry particle size is tracked in UCLALES-SALSA instead of ice crystal size and this lower size resolution seems to have an effect on ice crystal sedimentation. The other reason is related to the model dependent technical implementations such as the advection flux limiter method. Overall, the initial profiles in the presented case study are such that even a small decrease in LWP leads to decreased radiative cooling and turbulence, and this will prevent mixing of moisture from low altitude to cloud base. Thus, the model with default setup is more sensitive to ice formation close to the tipping point where technical details have largest impact.

Changes in the related paragraph in the revised manuscript:

Compared to the model results in Ovchinnikov et al. (2014), IWP in UCLALES-SALSA declines faster after the peak IWP has been reached in ICE4. One reason for this is that dry particle size is tracked in UCLALES-SALSA and this seem to have an important effect on ice crystal sedimentation. The other reason is related to the model dependent technical details. In Ovchinnikov et al. (2014) it was also stated that when the ice number concentration gets higher the differences between models are more caused by discrepancies in microphysics than cloud dynamics. This underlines the sensitivity and significance of microphysics.

**Major comment R2.4** How is droplet number concentration specified in the ISDAC ICE4 simulation? Is this prognostic? If so, it would be insightful to see the droplet number concentration in Figure 4. Is this why the results are so different than the intercomparison?

**Answer to R2.4** We have now added the droplet number concentration time series to Fig. 4. Droplet number concentration is prognostic in all fixed ice and prognostic ice simulations. Droplet number concentration decreases when ice number concentration is increasing but that is not the driving force behind complete removal of liquid phase, as explained below in the revised manuscript text. Additional explanation to this question is given also in the answer to the question R2.3.

Changes in the related paragraph in the revised manuscript:

Figure 4a shows that in the prognostic ice simulation LWP starts to increase after 4.5 hours of simulation. This is caused by a decrease of ice number concentration (Fig. 4c) to such a low level  which allows more water vapour for condensation to liquid droplets. The same figure also depicts how the ice number concentration is set to a target value (simulation ICE4) and how the concentration is stable until the cloud dissipates. Figure 4d depicts how droplet number concentration lowers especially right after spinup period when ice number concentration is increasing. However, changes in droplet number concentration is not the driving force behind complete removal of liquid phase. Figure 4e illustrates how the whole cloud with prognostic droplet freezing descends and how the ICE4 is affected by entrainment both below and above the cloud,cloud gets thinner and dissipates. In all simulations, droplet number concentration is specified as prognostic variable.

**Major comment R2.5** It is not clear how the artificial movement of aerosols between bins for numerical stability is affecting the results (lines 240-243).
**Answer to R2.5** The movement of aerosols between bins is a feature of the model that is needed for stability. This explains the numerical artefact seen in Fig. 6, but has no effect on the results. This is now stated in the manuscript.

Related part of the paragraph in the revised manuscript:

The increase in the total number of particles in bin 1 is a numerical artefact caused by the bin adjustment routine, which can move particles from one bin to another in order to keep the dry size within the predefined bin limits. When a large  proportion of particles in bin 2 are activated as cloud droplets, some of the remaining are moved to the smaller bin to avoid numerical problems. However, this numerical artefact does not affect the results.

**Major comment R2.6** Please explain why droplet freezing occurs throughout the cloud while for the same case Savre and Ekman (2015) found droplet freezing at cloud top. A more detailed discussion of how aerosols and droplet and ice crystal activation are represented in the two models is needed to understand why simulations in the two studies differ.
**Answer to R2.6** Upon closer inspection, we found that the comparison to Savre and Ekman (2015) was somewhat flawed and hence the comparison was removed from the manuscript. They used a different cloud case and this could explain the differences in droplet freezing profiles.
What happens within the cloud layer in our model is that, when supersaturation and cloud activation are explicitly modelled as in UCLALES-SALSA, unactivated particles can penetrate through the cloud layer in a down-draft with low supersaturation and later come back to the cloud with up-drafts and activate due to the higher supersaturation. Then freezing happens in the up-drafts throughout the cloud. If activation is not modelled with this level of details (any model,not just Savre and Ekman, 2015), activation and freezing might happen too early or late and in a wrong part of the cloud

Changes in the manuscript:

Figure 7c further illustrates an interesting behaviour of ice particle formation. In the beginning of the simulation ice particles are formed throughout the cloud, but later the most intensive formation takes place at the top of cloud where fresh  INPs are entrained into the cloud layer. However, the maximum supersaturation in these entraining downdrafts is so low, that only the

largest particles are able to form cloud droplets and consequently freeze. The smaller ones penetrate through the cloud layer as interstitial aerosol particles (i.e. unactivated particle), and are able to form cloud droplets (i.e. activate) and ice particles at the cloud base when they are recirculated back to the cloud with higher supersaturation. This can be well seen at the end of simulation as there is two peaks in vertical profile of freezing rate. Such phenomena can be only simulated with explicit calculation of in-cloud supersaturation and representation of aerosol size distribution and chemical composition like is done in UCLALES-SALSA. If activation is not modelled with this level of details, activation and freezing might happen too early or late and in a wrong part of the cloud. Overall, Figs. 6 and 7c nicely demonstrate how the relative  proportions of particles in different hydrometeors are size dependent and how sectional description for aerosols is required to be able to simulate such processes in LES models.

**Major comment R2.7** Lines 275-277: More details of the simulations are needed to understand whether this is a correct statement.
**Answer to R2.7** This comment has been addressed in answer to question R2.3

**Minor comment R2.8** Line 198: ". . ..concentration is was. . .". Please reword.
**Answer to R2.8** We rewrote the related paragraph.

**Minor comment R2.9** Line 202: "was adjusted"
**Answer to R2.9** Spelling corrected as suggested.

**Minor comment R2.10** Line 203: "represents"
**Answer to R2.10** Spelling corrected as suggested.

---

## Author Response (AR2)

**Response to the comments about the submitted paper: Modelling mixed-phase clouds with large-eddy model UCLALES-SALSA, Ref. ACP-2019-1182.**

Dear Editor, dear Reviewers, we would like to thank the Editorial Board for considering our paper for publication in ACP and the reviewers for their constructive comments. We have addressed all of them and modified the paper accordingly. Our detailed answers follow. Text from the 1st revised manuscript that has been removed in the 2nd revised manuscript is marked in red. New text in the 2nd revised manuscript is marked in blue.

**Answers to Reviewer 2**

**Major comment R2.1** Thank you to the authors for responding to my questions/concerns. I'm still concerned that the results presented are very limited. The model is designed to study the relative importance of aerosols with different chemical compositions but only the most basic studies were done (only dust can form ice). It is not a new result that allowing ice to precipitate will reduce the IWC and allow LWC to persist longer or that smaller aerosols that are not activated at cloud top can fall through the cloud layer and be transported into the cloud layer at cloud base. At the very least, a complete analysis of the freezing of smaller aerosols in supersaturated updrafts would provide support for this statement (lines 285-287).

**Answer to R2.1** We thank the referee for these comments and agree that the mentioned results are not new. However, in our study these results are gained with a detailed model where e.g. the chemical composition of the ice nucleating particles is also a prognostic variable. Aerosols in our study consist of ammonium bisulphate and dust where sulphate is needed for cloud activation and dust acts as INP. We have elaborated in our earlier answer to question R1.3 why we used only dust as INP in this study. As requested, we have now included a analysis of freezing of smaller hydrometeors in updrafts and downdrafts within the cloud layer. We added a figure to support this analysis. We also refined our references in the same chapter.

**Changes in the manuscript concerning R2.1**

Last chapter of Sect. 3.3 rewritten:

Figure 7c further illustrates an interesting behaviour of ice particle formation. In the beginning of the simulation ice particles are formed throughout the cloud, but later the most intensive formation takes place at the top of cloud where fresh INPs are entrained into the cloud layer. However, the maximum supersaturation in these entraining downdrafts is so low, that only the largest particles are able to form cloud droplets and consequently freeze. The smaller ones penetrate through the cloud layer as interstitial aerosol particles (i.e. unactivated particle), and are able to form cloud droplets (i.e. activate) and ice particles at the cloud base when they are recirculated back to the cloud with higher supersaturation. This can be  seen in Fig. 8. Figure 8a shows how in size bin 2 cloud droplets and ice particles are more frequent in updrafts compared to Fig. 8b which illustrates how aerosols are more favourable in downdrafts. Additionally, ice particles dominate in bigger sizes as aerosols freeze both in down- and updrafts (size bin 3 shown Figs. 8c and 8d). Simulated freezing in different vertical velocity conditions in other size bins does not differ from results shown already in Fig. 6. Lower peak at end of simulation  in vertical profile of freezing rate  in Fig. 7c also shows how recirculated aerosols are frozen in the cloud layer. Such phenomenon has been modelled before e.g. in Solomon et al. (2015), however here the cloud is simulated with explicit calculation of in-cloud supersaturation and representation of aerosol size distribution and chemical composition. If activation is not modelled with this level of details, activation and freezing might happen too early or late and in a wrong part of the cloud. Overall, Figs. 6 7c, and 8 nicely demonstrate how the relative proportions of particles in different hydrometeors are size dependent and how sectional description for aerosols is required to be able to simulate such processes in LES models.

**Major comment R2.2** This model is glaciating much earlier than all 11 models included in the ISDAC intercomparison for fixed ice number concentration. It would be useful to understand why in order to put the prognostic ice simulations into context. The authors say this is because the model is tracking the dry aerosol size. Why would this produce this result? Do the authors believe that this model is producing more realistic sensitivity that is not in the other models?

**Answer to R2.2** We chose to track aerosol, cloud and ice particle dry size to have information about the origin of the seed aerosol and the freezing cloud droplet. However, tracking dry size means reduced resolution for their hydrated size, especially as we do not assume any pre-defined shape for ice particle size distribution within bins. This is not an issue for aerosol or cloud droplets, but ice particles can be so large that their dry size is not anymore a good tracer for particle properties – this is one reason why rain drops are always described with their wet sizes. Low ice particle size resolution has an impact on ice depositional growth and especially sedimentation. Also, tracking dry size during particle sedimentation means that ice particles that have different hydrated sizes can be mixed. The same issue is related to the other mixing processes including advection and diffusion. So question is why we wanted to use the presented microphysical model setup. In case wet size is followed, we loose the information of original aerosol size distribution within ice particles. For example, Karlsson et al. show how residuals of ice crystals contain really small aerosol nuclei (Karlsson et al.: The role of nanoparticles in Arctic cloud formation, Atmos. Chem. Phys. Discuss., `https://doi.org/10.5194/acp-2020-417`, in review, 2020.). They are probably there due to secondary ice formation. The setup used in our simulations, i.e. tracking dry size, is currently the only possible way to describe how these small nuclei preserve in cloud processes. If microphysics follows the wet size, all this information about small nuclei is lost when bins following wet size are averaged by design. Tracking both wet and dry sizes correctly would be computationally really heavy with spectral microphysics.

The other explanation is related to model dependent technical details that are rarely changed or described in publications, but can have an effect on simulation results. One of these is advection flux limiter. By default UCLALES-SALSA uses minmod, but changing this to monotonized central (built-in options in UCLALES) will have a significant impact on results. However, it should be noted that this is not the only model setting that have an effect on results.

To show the impacts of these two, we show here results from ICE4 simulations (Fig. figure R2.2-1) where ice bins are described with particle size (identical with the rain drop size bins) and/or the default advection flux limiter minmod is changed to monotonized central (an example of a model dependent technical detail). Clearly, tracking ice particle size helps to maintain cloud liquid, and the combined effects of the size-tracking and flux limiter methods is even more effective in maintaining cloud liquid. More detailed analysis of the sedimentation flux shows that using dry size means reduced ice particle flux especially in the lowest 200 m leaving more particles there to evaporate. Evaporative cooling leads to a surface inversion which prevents the uplifting of moist near-surface air. This can be seen from profiles shown in Fig. figure R2.2-2. As such, the higher sensitivity to ice number concentration is partly related to the initial conditions.

Although these specific changes made UCLALES-SALSA less sensitive on ice number concentration, we cannot exclude other technical reasons. Because this explanation is highly technical and both case and model specific, we would like to keep this part of the manuscript text as brief as possible. We added relevant details to the manuscript and leave the rest here.

Tracking aerosol dry size instead of ice particle size seem to produce a higher sensitivity than is in the other ISDAC models. It is unlikely that this sensitivity or those from the other models are realistic, because they assume a constant ice number concentration. As already stated in the manuscript, prognostic ice simulations (with dry or wet size tracking) should be used for this.

[Figure]

Figure R2.2-1: Time series of ice (IWP) and liquid (LWP) water paths based on the default and modified model versions.

[Figure]

Figure R2.2-2: Profiles of total water mixing ratio, ice-liquid water potential temperature and latent heating rate due to evaporation of ice particles.

**Changes in the manuscript concerning R2.2**

Second to last chapter of Sect. 3.1:

Compared to the model results in Ovchinnikov et al. (2014), IWP in UCLALES-SALSA declines faster after the peak IWP has been reached in ICE4. One reason for this is that dry particle size is tracked in UCLALES-SALSA and this seem to have an important effect on ice crystal sedimentation. Namely, sedimentation velocities and particle mixing (flux divergency) are here calculated for the dry size bins rather than bins tracking ice particle size. This reduces particle flux especially in the lowest 200 m leaving more particles there to evaporate. Evaporative cooling leads to a surface inversion which prevents the mixing of moist surface air. As such, the higher sensitivity to INP concentration is partly related to the initial conditions of the ISDAC case study. The other reason is related to the model dependent technical details. Our test simulations (not shown) indicate that changing model options, such as flux limiter method, impact IWP and LWP so that the gap between UCLALES-SALSA and the other models decreases. In Ovchinnikov et al. (2014) it was also stated that when the ice number concentration gets higher the differences between models are more caused by discrepancies in microphysics than cloud dynamics. This underlines the sensitivity and significance of microphysics.

**Major comment R2.3** Are the aerosols tracked after they form ice? After the ice sublimates, are the aerosols added back into the population? It looks like this is not taken into account in Figure 1. Including this process will greatly impact aerosol concentrations below the cloud layer and entrainment of aerosols at cloud base.

**Answer to R2.3** Aerosols are tracked after they form ice. Also after ice sublimation the aerosol are added back into the population. We have now elaborated description of sublimation in the manuscript.

**Changes in the manuscript concerning R2.3**

1) Sublimation arrow from ice to aerosols added to Fig. 1.

2) In Sect. Model description:

The aerosol, cloud and ice crystal size distributions are discretised into the bins according to the dry aerosol diameter, whereas the rain droplet size distribution is defined by the wet diameter of the droplet. Identical 2a and 2b size bins are used for aerosol, cloud droplets and ice. Such parallel bins are useful for tracking aerosol development through cloud activation, freezing and sublimation. Prognostic variables for each bin include aerosol number and masses of all compounds (water, sulphate, dust, organic carbon, sea salt, nitrate, and ammonium).

Deposition of water, i.e. diffusion limited condensation or evaporation of water vapour, is defined for aerosol when relative humidity (RH) is over 98% and always for other hydrometeors. This is based on the analytical predictor of condensation (APC) scheme by Jacobson (2005) and implemented following Tonttila et al. (2017) (Eqs. 7 and 8). According to this definition, the particles compete for the available water vapour. For solids, the condensation equation does not require Kelvin or Raoult terms. If ice sublimates, the immersed aerosol nuclei are added back to the aerosol population.

**Minor comment R2.4** Figure 2 legend: "bulk" and "bin" should be switched.
**Answer to R2.4** Labels switched accordingly.

[revised manuscript text omitted]

---

## Author Response (AR3)

**Corrections to the manuscript**

Dear Editor,
We have corrected the colors of the lines in the Figs. 6 and 8 as suggested.

Text from the 2nd revised manuscript that has been removed in the 3rd revised manuscript is marked in red. New text in the 3rd revised manuscript is marked in blue.

[revised manuscript text omitted]